# Advances in Understanding of the Immune Response to Mycobacterial Pathogens and Vaccines through Use of Cattle and *Mycobacterium avium* subsp. *paratuberculosis* as a Prototypic Mycobacterial Pathogen

**DOI:** 10.3390/vaccines9101085

**Published:** 2021-09-26

**Authors:** William C. Davis, Gaber S. Abdellrazeq, Asmaa H. Mahmoud, Kun-Taek Park, Mahmoud M. Elnaggar, Gaetano Donofrio, Victoria Hulubei, Lindsay M. Fry

**Affiliations:** 1Department of Veterinary Microbiology and Pathology, Washington State University, Pullman, WA 99164, USA; gaber.abdellatif@alexu.edu.eg (G.S.A.); Asmaa.mahmoud@wsu.edu (A.H.M.); Mahmoud.elnaggar@alexu.edu.eg (M.M.E.); V.hulebei@ws.edu (V.H.); lfry@wsu.edu (L.M.F.); 2Department of Microbiology, Faculty of Veterinary Medicine, Alexandria University, Alexandria 22758, Egypt; 3Veterinary Quarantine of Alexandria, General Organization for Veterinary Services, Ministry of Agriculture and Land Reclamation, Dokki, Giza 12611, Egypt; 4Department of Biotechnology, Inje University, Injero 197, Kimhae-si 50834, Korea; pkt9138@hotmail.com; 5Department of Medical-Veterinary Science, University of Parma, 43126 Parma, Italy; gaetano.donofrio@unipr.it; 6Animal Disease Research Unit, USDA-ARS, Pullman, WA 99164, USA

**Keywords:** paratuberculosis, tuberculosis, vaccines, CD8 cytotoxic T cells, bovine

## Abstract

Lack of understanding of the immune response to mycobacterial pathogens has impeded progress in development of vaccines. Infection leads to development of an immune response that controls infection but is unable to eliminate the pathogen, resulting in a persistent infection. Although this puzzle remains to be solved, progress has been made using cattle as a model species to study the immune response to a prototypic mycobacterium, *Mycobacterium a. paratuberculosis* (*Map*). As chronicled in the review, incremental advances in characterizing the immune response to mycobacteria during the last 30 years with increases in information on the evolution of mycobacteria and *relA*, a gene regulating the stringent response, have brought us closer to an answer. We provide a brief overview of how mycobacterial pathogens were introduced into cattle during the transition of humankind to nomadic pastoralists who domesticated animals for food and farming. We summarize what is known about speciation of mycobacteria since the discovery of *Mybacterium tuberculsis* *Mtb*, *M. bovis* *Mbv*, and *Map* as zoonotic pathogens and discuss the challenges inherent in the development of vaccines to mycobacteria. We then describe how cattle were used to characterize the immune response to a prototypic mycobacterial pathogen and development of novel candidate vaccines.

## 1. Introduction

Comparative phylogenomic studies show that *Mycobacterium tuberculosis* (*Mtb*), *M. bovis* (*Mbv*), and *M. avium* subsp. *paratuberculosis* (*Map*) are members of a lineage of bacteria that emerged well before the dawn of civilization [1,2]. Their emergence as pathogens of humans and livestock is associated with a change in living conditions that occurred during the transition by humans from hunter gatherers to nomadic pastoralists that began to domesticate animals for use in food and farming in the Mediterranean basin about ≈10,000 years ago [3,4,5]. The establishment of settlements brought humans and their domesticated animals into closer association and increased the risk for cross-exposure and spread of mycobacterial pathogens [6]. The actual demonstration that these mycobacteria were the causative agents of tuberculosis and paratuberculosis occurred during the last years of the 19th century [7,8]. In the ensuing years, information has been obtained showing mycobacteria have continued to evolve, giving rise to new lineages causing disease in humans and domesticated farm animals (reviewed in [9,10]). Spillover has occurred, leading to establishment of lineages in wildlife [1,2,11,12,13]. These revelations have emphasized the need to develop a global approach to studying the immune response to mycobacterial pathogens and a common strategy for vaccine development.

A central component of such a strategy requires elucidation of how mycobacterial pathogens dysregulate the immune response, allowing for establishment of a persistent infection. The ancestral mycobacterium that developed the ability to survive in the vertebrate host acquired genes that modulate the immune response, allowing for establishment of a persistent infection. These genes have been retained during speciation. Clues to their identification and how they modulate the immune response have proven elusive. As discussed in this review, we conducted studies over the past 30 plus years focused on (1) development and use of monoclonal antibody (mAb) reagents to characterize the immune system of cattle [14,15,16,17,18] and (2) development of cattle as a model large animal species to conduct comparative studies on the immune response to mycobacterial and other pathogens and vaccine development. These cumulative studies have brought us closer to identifying the genes involved in modulation of the immune response by mycobacterial pathogens and to development of novel candidate vaccines against *Map* and *Mbv.* These studies also revealed the importance of the CD8 cytotoxic T cell response to mycobacterial pathogens in protective immunity.

## 2. *Mycobacterium avium* subsp. *paratuberculosis*, a Prototypic Mycobacterial Pathogen

At the initiation of our studies, investigations on the immune response to mycobacterial pathogens were primarily focused on analysis of the immune response to *Mtb*, using mice as the model species for research (reviewed in [19,20]). Limited information was available on the immune response to *Map*. It was known at the time that *Map* was the causative agent of paratuberculosis (Johne’s disease, JD) [7]. The potential of *Map* as a zoonotic pathogen had been reported [21]. Due in large part to evidence that *Map* was becoming a major disease problem in the U.S. dairy industry, as well as stakeholder and veterinarian requests, the USDA commissioned the National Research Council of the National Academy of Sciences to establish a committee on the ‘Diagnosis and Control of Johne’s Disease’ [20]. The committee was instructed to conduct a thorough review of past research on JD in domestic and wild ruminants regarding methods of diagnosis, the mode of transmission, current programs for control and prevention of *Map* infection, and a review of the relationship of JD in ruminants with Crohn’s disease (CD) in humans. Significant knowledge gaps were identified in all areas. A series of recommendations were made for the conduct of research that included: (1) clarification of whether there is an age-related difference in susceptibility to infection; (2) development of better methods to identify animals at the early stages of infection; (3) completion of the *Map* genome sequencing project, which is in progress at the USDA National Animal Disease Center (NADC); (4) acceleration of research on development of better diagnostics and vaccines; (5) conducting research on the immune response to *Map* and development of ways to modulate the immune response to elicit protection; (6) exploration of the feasibility of developing recombinant vaccine technology for development of a vaccine that elicits a protective immune response; and (7) conducting further research to establish whether *Map* is a zoonotic pathogen.

As a result of the recommendations of the committee, a multi-institutional program, the Johne’s Disease Integrated Program in Research (JDIP), was funded by the USDA. The program provided funding and an opportunity for investigators to meet and initiate collaborations. Participation in the program provided an opportunity to continue our research and focus on filling in the gaps in knowledge on the immune response against *Map* and use the information to develop vaccines.

## 3. Immune Response to *Map*

One of the recommendations made by the committee on ‘Diagnosis and Control of Johne’s Disease’ was to determine whether there is an age-related difference in susceptibility to infection with *Map*. Suggestions that there might be a difference were based on field observations and trial experiments of limited duration. In addition, information regarding the composition of the immune system in cattle, as well as reagents to study the bovine cell-mediated immune response to *Map*, were limited (reviewed in Rideout et al. [20]). Methods available at the time the first studies were conducted were indirect, and failed to answer the question [22]. Further studies were conducted later in collaboration with investigators at the USDA ARS NADC. Neonatal calves were used in the first studies. Tritiated thymidine incorporation was still used as the primary method of analysis of the proliferative response to antigen (Ag) stimulation ex vivo in tissue culture [23]. Flow cytometry was introduced as a method to characterize the phenotype of cells present in cultures of PBMC stimulated with PPD or sonicates of *Map* (SAg) [24]. Analysis revealed all animals were infected when exposed under experimental conditions and developed an antibody response and a cell-mediated response. Analysis of the recall response revealed there was a proliferative response to stimulation with PPD and soluble antigens (Sags) of *Map.* Analysis of the phenotype of cells proliferating in response to Ag stimulation revealed memory CD45R0^+^ CD4 T cells were the major component of this population [23]. A follow-up study yielded similar results and revealed that memory CD45R0^+^ CD8 T cells were detected, but CD4 T cells were dominant in all cultures [24].

## 4. Early Events of Infection through Direct Infection of the Ileum

Two methods were used to gain further information on the early events of infection and the immune response to *Map*: direct injection of *Map* into the ileum via surgical intervention [25] and introduction through a cannula [26] (Figure 1). Tissues from directly inoculated calves were collected at intervals over nine months post-inoculation (PI). These studies revealed *Map* rapidly disseminates from the ileum to the mesenteric lymph nodes. Discernable lesions were not detected until six months post-infection. Flow cytometric analysis of the recall response at 8 and 9 months PI revealed infection led to development of a proliferative response to PPD dominated by CD4 T cells. The response of CD8 T cells was less pronounced [25].

Placement of a cannula in the ileum provided continuous access to the ileum and avoided the need for multiple invasive surgeries [26]. In addition, it provided a method that could be used to study the progressive changes in pathology associated with development of ileitis. The use of flow cytometry to study the immune response to *Map* was refined. Initial studies revealed flow cytometry could replace use of tritiated thymidine to study the proliferative response to Ag stimulation. Antigen-specific CD4 and CD8 T cells proliferating in response to stimulation with Ags in vitro could be distinguished by an increase in cell size and expression of the memory T cell marker CD45R0, as well as molecules upregulated only on activated cells proliferating in response to Ag stimulation [26]. Endoscopic examination of the ileum following direct introduction of *Map* revealed no gross detectable lesions during the duration of the study. Flow cytometric analysis of the recall response demonstrated direct introduction of *Map* into the ileum elicited a similar proliferative recall response dominated by CD4 T cells. Analysis of lymphocytes isolated from the jejunum and ileum at the end of the study revealed limited difference in the activation status of cells obtained from control calves and calves inoculated with *Map*.

## 5. Original Isolates of *Map* from Humans with CD Elicited an Immune Response Similar to the Response Elicited by Isolates from Cattle

The development of the cannulated ileum model provided an opportunity to obtain information that might provide insight on whether *Map* is a zoonotic pathogen. As reviewed in Rideout et al., insufficient information had been obtained to establish whether *Map* is the causative agent of CD [20]. A study was conducted with the original isolates of *Map* obtained from patients with CD to determine if passage through humans altered their immunogenicity or ability to induce pathological changes characteristic of changes observed with isolates from cattle [27]. The study showed there were no clear difference in changes in the ileum associated with direct inoculation of the ileum compared with the previous studies with an isolate of *Map* obtained from cattle. Biopsies showed the bacteria were rapidly cleared from the ileum with no evidence of colonization throughout the 11 months of study. Serial analysis of the recall response to SAg and PPD showed the early response was similar to the response observed with a bovine isolate, a response dominated by CD4 T cells. The CD8 T cell response was less at the early stages of infection but increased by 11 months after infection. Quantitative RT-PCR of IFN-γ, IL-17, and IL-22 with cells isolated from the ileocecal lymph node and mesenteric lymph node of experimentally infected calves and the naturally infected cows showed a similar increase in expression.

The studies conducted during this timeframe demonstrated exposure to *Map* leads to infection and the development of an immune response that controls, but does not clear, the infection. After a variable period of time, there is a breakdown in this type of immunity leading to clinical disease. Use of isolates of *Map* from humans resulted in development of a latent infection similar to the latent infection that occurs with isolates from cattle, supporting findings indicating *Map* is a zoonotic pathogen. Extensive studies have been reported since the publication of these findings that show *Map* is the causative agent of CD. See references [28,29,30] for more detailed information.

## 6. Site-Directed Mutagenesis

Having access to the ileum presented an opportunity to study the immune response to *Map* during the early and late stages of infection. However, to make good use of the methodology and the ability to use flow cytometry to characterize the phenotype of cells involved in the immune response to *Map*, there was a need to increase understanding of the genetic basis of virulence and mechanisms used by *Map* to establish a persistent infection. The sequencing of the genome of *Map* was completed concurrent with our initial studies. Transposon mutagenesis had been adapted for use in the study *Map* by others in an attempt to identify genes associated with virulence and modulation of the immune response [31,32,33]. Methods had also been developed to use site-directed mutagenesis for selective deletion of genes in *Mtb* and other slow-growing mycobacteria [34]. We adapted the latter method for use with *Map* [35] and selected two genes for deletion, on the basis of information suggesting that they might affect virulence and modulate the immune response to *Map*: protein kinase G (*PknG*), a gene encoding an enzyme secreted by mycobacteria following entrance into lysosomes in macrophages, and *relA*, a gene involved in regulating the stringent response (the ability to survive in unfavorable environments) [36]. The JDIP had organized a collaborative study to compare the properties of mutants developed in different laboratories. A three-phase strategy was developed to identify the best candidates to test for their capacity to elicit a protective immune response in goats as a representative ruminant species (reviewed in [37]). A reduction in survival of *Map* in macrophages was used as the first criterion for further testing of the potential efficacy of *Map* mutants in mice. A reduction in tissue colonization of mice was used for the final selection of candidate mutants for testing in goats. Neither of our mutants exhibited reduced survival in macrophages, leading to their exclusion from further testing. Testing of the final candidates demonstrated none of the mutants stopped fecal shedding. One reduced the lesion score (reviewed in [37]). The immune response to the candidate mutant vaccines was not characterized in mice or goats.

Although our mutants were excluded from further evaluation, studies on the genes we selected for deletion, PknG and *relA*, had suggested they were good candidates for developing attenuated vaccines. Studies by Walburger et al. found that deletion of *PknG* disrupted the ability of bacillus Calmette–Guérin *Mycobacterium bovis* (BCG) to survive in macrophages in an ex vivo model system [38]. Further studies demonstrated chemical blockade of *PknG* secreted into macrophages by BCG and *M. tuberculosis* (*Mtb*) allowed for lysosome–phagosome fusion to occur, resulting in killing of intracellular bacteria. These data suggested this might be one of the mechanisms used by mycobacteria to establish a persistent infection.

A report on the study of the role of *relA* in regulating the survival of *Mtb* (H37Rv) in unfavorable environments in mice revealed deletion of *relA* affected survival in vivo [39]. Comparison of granuloma formation in the lungs elicited by H37Rv with a deletion in *relA* revealed granulomas developed and then began to resolve by the end of the study. Bacteria were not detected in residual granulomas. The data suggested the molecular pathways regulated by *relA* might be involved in creating an intracellular environment that allowed for establishment of persistent infection. Studies were conducted in cattle and goats to determine whether the same effect occurs with other mycobacterial pathogens. Comparison of survival of wild-type *Map* with a *Map/PknG* deletion mutant in calves and goats revealed deletion of *PknG* did not prevent establisment of a persistent infection. However, colonization of tissues was reduced in comparison with wild-type *Map*, indicating deletion did have an effect on survival [40]. In contrast, comparison of survival of a *Map/relA* deletion mutant with wildtype *Map* revealed deletion prevented establishment of persistent infection. No bacteria were detected, by culture or PCR, within nine different tissues collected from both calves and goats at necropsy [40], duplicating the observations made by Dahl et al. in mice [39]. The data suggested the reason why the mutant could not establish a persistent infection was that an immune response developed that cleared the infection.

## 7. Analysis of Immune Response Elicited by *Map/relA*

Although intense investigations were underway by others to determine the role of *relA* and associated genes in regulating the capacity of bacteria, including mycobacteria, to survive in unfavorable environments, referred to as the stringent response [36,41], the investigations did not include studies to determine whether the molecular pathways regulated by *relA* might be involved in dysregulating the immune response and facilitating establishment of persistent infections. This was a surprising omission in the investigation of *relA* in mycobacteria and other bacteria. The studies by Dahl et al. [39] and our follow up studies suggested genes under the reglation of *relA* did play a role in modulating the immune response. Methods were developed to explore this possibility. An ex vivo tissue culture platform was developed to study the recall response elicited by vaccination of cattle with the *Map/relA* deletion mutant. This included: (1) development of a monoclonal antibody to CD209 uniquely expressed on antigen-presenting cells (APC) (i.e., blood-derived dendritic cells (bDC) and monocyte-derived dendritic cells (moDC)) in order to study the proliferative response to antigens (Ag) processed and presented by APC to CD4 and CD8 T cells [42,43], and (2) use of a flow cytometric assay to determine the phenotype of lymphocyte subsets proliferating in response to antigenic epitopes of Ags processed and presented by Ag-primed APC. Analysis of the recall of the response with PBMC and PBMC depleted of monocytes (mdPBMC) from a steer vaccinated with the *Map/relA* mutant revealed CD4 and CD8 T cells proliferated in response to Ag processed and presented by APC pulsed with *Map/relA* [43]. This demonstrated the inability of *Map/relA* to establish a persistent infection could be attributed to the development of an immunre response that cleared infection.

## 8. Identification of a Target of the Immune Response Elicited by *Map/relA*

To gain further information on the immune response to *Map/relA*, studies were conducted to determine the target of the immune response. Multiple *Map* Ags had been previously identified as targets of the antibody response to *Map* [44]. One protein of particular interest was a 35 kD membrane protein, MMP, encoded by MAP2121c. Previous studies had shown it played a role in invasion of epithelial cells [45]. Importantly, the protein had been expressed and was provided by Bannantine for collaborative studies. Comparison of the recall proliferative response to APC primed with *Map/relA* or MMP demonstrated comparable CD4 and CD8 T cell responses were elicited by APC primed with either *Map/relA* or MMP. Further analysis demonstrated the response was MHC-restricted [43,46].

## 9. Analysis of the Effector Activity of CD4 and CD8 T Cells Proliferating in Response to Stimulation with APC Pulsed with *Map/relA* and MMP

Investigations to this time point established *Map/relA* elicited an immune response that could be studied ex vivo, and that a target of the response was a 35 kD membrane protein, MMP. The studies had not demonstrated whether the cells proliferating in response to stimulation with APC pulsed with *Map/relA* or MMP had any direct effect on the viability of bacteria present in macrophages used as targets. Additional methods were developed to extend studies on the immune response to *Map/relA* and MMP, as well as to characterize the effector activity of the CD4 and CD8 T cells proliferating in response to stimulation with Ag-primed APC. The first method was based on studies by Worku and Hoft that demonstrated macrophages infected with BCG could be used as target cells to study the effector activity of responding lymphocyte subsets from human volunteers vaccinared with BCG [47]. Comparison of BCG growth inhibition in infected macrophages overlayed with cultures of rested or BCG-stimulated PBMC from immunized volunteers demonstrated replication of BCG was arrested in the presence of autologous BCG-stimulated PBMC. As described by Park et al., a recall response could be elicited with one round of stimulation of PBMC from a vaccinated steer with Ag-primed APC (Park et al., 2016). Two rounds of stimulation with Ag-primed APC were needed with PBMC from an unvaccinated steer to generate enough cells for analysis of the primary immune response to Ag stimulation ex vivo [46] (Figure 2).

The second method was designed to overcome the limitations in the use of the colony-forming unit assay (CFU), which requires six or more weeks of culture before colonies of bacteria are large enough to count. Concurrent with our studies, Nocker et al. conducted studies to improve on methods available for identifying and quantifying the concentration of live bacteria present in preparations of bacteria obtained from various sources [48,49]. Propidium monoazide (PMA), a fluorescent dye taken up by DNA, was identified for potential use in distingishing live bacteria from dead bacteria in mixed preparations of bacteria. One of their studies was focused on developing an assay to replace the CFU assay for quantitation of live *Map* in foods and other biological sources. Aside from the bacterium being slow-growing, the tendency to clump made the CFU assay less accurate for assesing the concentration of bacteria in isolates. Kralik et al. demonstrated that a single copy gene, F57, could be used with quantitative PCR to determine the concentration of live bacteria present in a mixed population of bacteria [50] (Figure 3).

The method was adapted to quantify the number of viable bacteria present in infected macrophage used as target cells before and after mixing with PBMC from a *Map/relA*-vaccinated steer [46,52]. Analysis revealed the apparent arrest of growth of BCG present in target cells observerd by Worku was attributed to killing of intracellular bacteria by CD8 cytotoxice T cells and not to an arrest of growth. Killing of intracellular *Map* was evident by six hours following incubation of *Map*-infected target cells with PBMC from a vaccinated steer stimulated with APC primed with *Map/relA*. Comparative studies of the recall response revealed stimulation of PBMC with APC primed with either *Map/relA* or MMP elicited an identical proliferation of CD8 CTL. Stimulation always elicited a comparable proliferative response by CD4 T cells. Similar results were obtained with PBMC from naïve steers stimulated twice with APC primed with either *Map/relA* or MMP. Analysis of the mechanisms used in killing intracellular bacteria revealed killing was mediated through the perforin-Granzyme B pathway [46].

## 10. Simultaneous Recognition of Antigens Processed and Presented by APC Is Essential for Development of CD8 CTL

A consistent observation throughout studies of the immune response to *Map* was that exposure to *Map*, *Map/relA*, PPD, *Map*-derived soluble antigen extract (SAg), or MMP always elicited a simultaneous proliferative response of CD4 and CD8 T cells. As with ongoing studies of the immune response to *Mtb* and *Mbv* BCG, the findings indicated CD4 T cells were serving as helper cells in development of CD8 CTL. It was unknown whether cognate recognition of Ags processed and presented by Ag primed APC was essential for development of CD8 CTL. Demonstration that CTL could be elicited ex vivo with a single peptide, MMP, provided an opportunity to answer this question with mdPBMC from *Map* free steers. Comparison of the proliferative response to MMP-primed APC with unseparated mdPBMC with mdPBMC depleted of either CD4 or CD8 revealed the proliferative response was greatly reduced in cultures depleted of CD4 or CD8 T cells. This provided evidence that both CD4 and CD8 T cells must be present at the time of Ag presentation to elicit development of CTL. However, the design of the experiments did not reveal how and when CD4 T cell help had to be delivered to elicit development of CD8 CTL. A study was conducted to duplicate the initial results but with a focus on examining the events occurring at the time of Ag presentation by MMP-primed APC. Monoclonal antibodies specific for MHC class I and class II (HLA-DR and HLA-DQ orthologues) were used to affirm the immune response was MHC-restricted [14,53]. As observed in the previous studies, both CD4 and CD8 T cells proliferated in response to stimulation with MMP-primed APC with CD8 developing CTL activity. CD8 T cells did not proliferate in the absence of CD4 T cells. Moreover, proliferation and development of CTL activity did not occur in cultures containing mAbs specific for MHC class I and class II molecules or cultures containing MHC class I alone or MHC class II alone. These studies demonstrated interference with Ag presentation and signaling through MHC I or MHC II was sufficient to block the proliferative response of both CD4 and CD8 T cells. It also demonstrated the same APC had to deliver peptide sequences derived from the same peptide at the same time for development of CD8 T cells [54] (Figure 4).

## 11. Candidate Vaccines for *Map* and *Mbv*

The studies with the *Map/relA* mutant provided information that suggested the genes involved in disrupting the immune response and allowing for establishment of a persistent infection are regulated through a gene common to mycobacteria and other bacterial pathogens. It was not possible to conduct studies to identify the genes at this stage of our invesitigation. However, the information obtained was sufficient to suggest that deletion might be all that is needed for developing *relA* deletion mutants that elicit a protective immune response. Studies with the MMP peptide suggested it might be sufficient for developing peptide-based vaccines for paratuberculosis. Two lines of investigation were initiated to explore each of these possibilities: (1) to determine the effect of deleting *relA* in another mycobacterial pathogen, and (2) to develop a delivery method and subsequent testing strategy to assess the efficacy of candidate vaccine peptides using MMP as a prototype Ag with known ability to elicit CTL activity.

## 12. *relA* Deletion Mutants

Studies in mice and cattle demonstrated deletion of *relA* makes mutants susceptible to immune elimination. Analysis in cattle demonstrated vaccination with *Map/relA* leads to development of CD8 cytotoxic T cells with the ability to kill intracellular bacteria. The data suggested *relA* may be the Achilles’ heel for mycobacterial pathogens and a gateway for developing effective attenuated vaccines for mycobacterial pathogens. To gain further information on this possibility, and on the potential use of *relA* deletion mutants as attenuated live vaccines, researchers developed a BCG *relA* deletion mutant for comparative studies. BCG was chosen to allow for differentiation between the immune response to BCG and that elicited by *Mbv*. It was also chosen because studies in humans and cattle had shown further modifications were needed to improve the efficacy of BCG as a vaccine. Although deletion of the RD1 gene complex reduced the virulence of *Mbv* BCG, it could still establish a persistent infection and elicit an immune response sufficient to control infection [55,56]. From studies with the H37Rv *relA* deletion mutant and *Map/relA*, it could be inferred that deletion of *relA* in BCG would increase susceptibility to immune elimination by CTL. Studies were conducted to compare the immune response to BCG and BCG/*relA* ex vivo and attempt to identify any differences in the immune response that would reveal how deletion of *relA* abrogates the capacity to establish a persistent infection. The studies included: (1) comparing the proliferative response elicited by live and heat killed bacteria; (2) the development of CTL; (3) expression of TNF-α, IFN-γ, and IL-17A by CD4 and CD8 T cells; and (4) MHC restriction of the immune response. The studies revealed both BCG/*relA* and BCG elicited comparable proliferative responses similar the responses elicited by *Map/relA*. No difference was detected in the capacity of BCG and BCG/reA to elicit development of CD8 CTL with the ability to kill intracellular bacteria. Initiation of killing was immediate as detected macroscopically with the detachment of adherent infected target cells following mixing with mdPBMC containing CD8 CTL. In contrast, the proliferative response was greatly reduced in cultures of mdPBMC stimulated with heat-killed bacteria. Little or no killing was detectable in cultures of infected target cells mixed with mdPBMC stimulated with heat-killed bacteria. Analysis of the mechanisms of killing revealed killing was mediated through the perforin, Granzyme B, granulysin pathway. Analysis of MHC restriction revealed the proliferative response and development of CD8 CTL was blocked in the presence of mAbs specific for either MHC I or MHC II. The main observed difference was in expression of TNF-α, IFN-γ, and IL-17 in cultures of PBMC stimulated with BCG or BCG/*relA*. Expression was higher in PBMC stimulated with BCG/*relA* [51]. Of particular interest was the fact that all three cytokines were secreted by CD4 and CD8 T cells.

## 13. Peptide-Based Vaccines

Development of peptide-based vaccines has proven difficult, especially for the design of vaccines that elicit a CD8 cytotoxic T cell response. The requirement for CD4 T cell help has been documented with different model systems (reviewed most recently in Laidlaw et al. (2016)). However, none of the model systems clearly established how and when CD4 T cell help is delivered to elicit development of CD8 CTL. Analysis of the requirement for eliciting CTL against *Map* provided an answer to this question. As reviewed here, studies with MMP demonstrated CD4 T cell help must be delivered simultaneously at the time of Ag presentation to CD8 T cells by APC primed with a peptide antigen [54]. Examination of the CD4 and CD8 T cell response to MMP processed and presented by APC demonstrated presentation always elicited a robust proliferative response. The proliferative response was blocked in the presence of antibodies specific for MHC class I and class II molecules and also blocked in the presence of antibody to either MHC I or MHC II alone. The observations revealed that CD4 and CD8 T cell target antigenic epitopes must be present in the same candidate vaccine peptides for processing and presentation to occur simultaneously and for signaling to occur from the APC and between CD4 and CD8 T cells [54].

## 14. Method of Delivery of Peptide-Based Vaccines

Various approaches have been examined to determine the best approach for delivery of peptide-based vaccines [57]. The approaches examined for MMP included: (1) packaging of MMP into nanoparticles for direct delivery as a vaccine and (2) optimization of the gene encoding MMP for incorporation into a shuttle vector for delivery by a viral vector. A commercially available nanoparticle vector comprised of poly(D, L-lactide-co-glycolide) and monophosphoryl lipid A (PLGA/MPLA) was tested for use as a vector for MMP [58]. Preliminary studies demonstrated a CTL response was elicited with APC pulsed with MMP incorporated into PLGA/MPLA [58]. Further studies demonstrated development of a standardized method of producing a nanoparticle–MMP complex with known activity was not possible. Therefore, efforts were redirected to examining the potential of developing a viral vector for delivery of MMP as a vaccine. Of critical importance for developing a viral-vectored vaccine is that any modifications of the gene encoding the candidate peptide, needed for expression in a viral vector, must not alter the immunogenicity of the native peptide. The gene encoding MMP was optimized for expression in mammalian cells, placed in a mammalian expression vector, and expressed to determine if immunogenicity was retained [59]. Comparison of immunogenicity of MMP expressed in *E. coli* (the form used in initial studies) with MMP modified for expression in mammalian cells demonstrated the modified peptide retained its ability to elicit CTL. This finding and the successful delivery of the secreted MMP, modified for expression in mammalian cells by an integrative third generation replicating incompetent lentiviral vector [59], indicate the MMP optimized expression cassette is ready to be placed in a non-integrative viral vector for testing as vaccine. Recombinant non-integrative viral vectors are an attractive class of vaccine delivery systems as they are safe and simple to construct and manufacture. Different classes of viruses have been tested as viral vectors (vaccinia virus (VV), vesicular stomatitis virus (VSV), adenovirus (AdV), adeno-associated virus (AAV), poxvirus (ALVAC), bovine herpes virus 4 (BoHV-4), and others [60]). Each vector presents particular advantages and disadvantages, depending on their biological characteristics and on the host that needs to be protected against a specific pathogen. At present, it is difficult to predict which viral vector will be the best to use to with MMP. A specific virus vector should be able to confer selective immunization only against a specific pathogen and not toward others. Our current efforts are focused on use of the BoVH4 vector for examining the efficacy of MMP as a peptide-based vaccine for paratuberculosis.

## 15. Summary

Development of the bovine model for the study of mycobacterial pathogens has provided opportunity to use a prototypic mycobacterial pathogen to advance our understanding of the immune response to mycobacterial pathogens and determine whether there is an Achilles’ heel common to all mycobacterial pathogens that will enable development of protective vaccines. The development of the bovine model has introduced a platform where information on the immune response to mycobacteria and candidate vaccines obtained from studies in mice, humans, non-human primates, and cattle can be compared. Of special importance, the platform provides an opportunity to elucidate the signaling that occurs between APC, CD4, and CD8 T cells, leading to the development of a cell-mediated immune response. The development of CD8 cytotoxic T cell activity can be studied in detail. Due to the size and longevity of cattle, multiple studies can be conducted with a single animal, keeping the genetic background constant, similar to the use of an inbred strain of mice. Multiple animals can be used when MHC restriction needs to be considered as a variable. The methods developed to study the immune response ex vivo in cattle can be used to duplicate studies with cells from all three species. Added advantages to the use of cattle as an outbred species model are that cattle are phylogenetically more closely related to humans, with the lineages of genes regulating the immune response identical to those in humans. Moreover, cattle are one of the natural host species of *Mtb*, *Mbv*, and *Map*.

Thus far, use of the bovine model has provided an opportunity to begin sorting out the features of the immune response to mycobacterial pathogens reported in the extensive studies conducted on *Mtb* and BCG during the past years. This has included obtaining data showing there is a gene that has been conserved during speciation of bacteria that may be the Achilles’ heel for mycobacterial pathogens, as well as data showing macrophages are not a safe haven for mycobacterial pathogens. relA regulates expression and function of genes essential for maintenance of a persistent infection. Cytotoxic CD8 T cells kill intracellular bacteria using the perforin, Grn B, granulysin pathway. It has also included obtaining data revealing how and when CD4 T cell help is delivered to elicit development of CD8 cytotoxic T cells. T cell help must be delivered concurrent with antigen presentation to CD4 and CD8 T cells by APC primed with antigen. The data indicate that the peptide antigens used to elicit cytotoxic T cells must contain epitopes for presentation through MHC class I and class II molecules.

It remains to be determined whether deletion of *relA* is sufficient for development of attenuated vaccines or is just another step forward and opportunity to determine how mycobacterial pathogens modulate the immune response to evade immune elimination. The cumulative studies have suggested the genes and gene products involved in modulating the immune response have been conserved through speciation of bacteria. *relA*, a global regulator gene, and related homologues, modulate the expression of (p) ppG, an alarmone that regulates the expression of multiple genes involved in survival of bacteria in unfavorable environments. It is implicated in playing a role in virulence in multiple lineages of bacteria, including mycobacteria (reviewed in [61]). A single copy of a *relA Mbv* homologue, *relA_Mtb_*, is encoded in all mycobacterial pathogens. Surprisingly, little attention has been directed in these studies toward the possibility that loss of ability to survive in vivo in absence of *relA* might be attributable to development of an immune response that facilitates clearance of infection. Both *Map* and BCG with an intact *relA* and deletion mutants elicit the development of cytotoxic T cells. Comparison of the immune response has not revealed a clear difference. Likewise, further studies are needed to determine if the CTL response elicited by a virus vectored 35 kD candidate peptide vaccine is sufficient to prevent establishment of a persistent infection by *Map*, or if it is the first step in developing a multi-peptide-based vaccine. It is now evident that each peptide considered for inclusion in a vaccine must contain epitopes for simultaneous presentation by MHC class I and II molecules.

## Figures and Tables

**Figure 1 vaccines-09-01085-f001:**
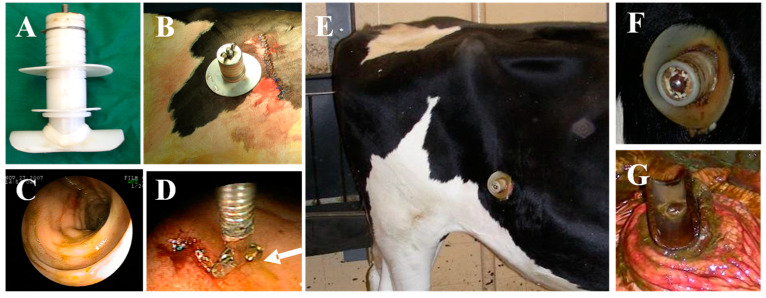
(**A**) Pictures showing the modified cannula used to cannulate calves before surgery. (**B**) Appearance of cannula after placement in the ileum. (**C**) Appearance of ileum before infection with *Map*. (**D**) Collection of a pinch biopsy. (**E**,**F**) Pictures showing the status of the cannula 8 months after surgery. Inflammation at the site of the implantation was kept at a minimum by keeping the site clean. (**G**) Inspection of internal portion of the cannula at necropsy showed minimal inflammation around the cannula [26].

**Figure 2 vaccines-09-01085-f002:**
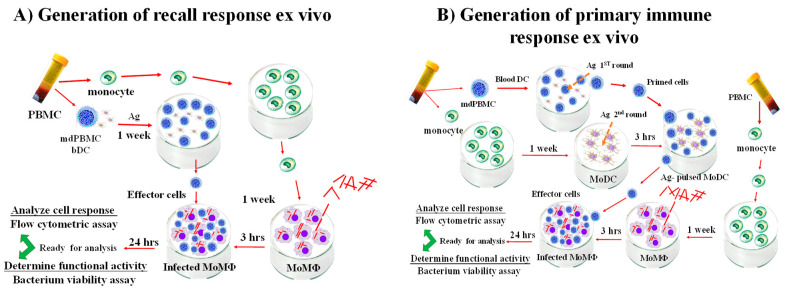
Flow diagrams illustrating the platforms developed to study the recall and primary cellular immune responses to live and peptide-based vaccines ex vivo. (**A**) For analysis of the recall response, PBMC from vaccinated cattle are separated into monocytes and mononuclear cells containing CD4 and CD8 T cells, γδ T cells, and CD209 positive blood dendritic cells (bDC). The bDC are stimulated directly without separation and cultured for a week. The monocytes are cultured in parallel for a week to generate monocyte-derived macrophage (MoMΦ) target cells. At one week, the MoMΦ are infected with *Map* or BCG fpr 3 h, washed to remove free bacteria, and then overlayed with stimulated or unstimulated monocyte depleted PBMC (mdPBMC). At 24 h post-stimulation, one set of cells are processed for flow cytometric analysis. The remaining set of cells are processed to isolate DNA from bacteria to determine the extent of killing mediated by CD8 cytotoxic T cells using a bacterium viability assay. (**B**) For analysis of the primary immune response, PBMC from naïve cattle are separated into mdPBMC containing bDC and monocytes. The mdPBMC are stimulated with Ag for one week. The monocytes are cultured with granulocyte/monocyte-stimulating factor (GMCSF) and interluekin 4 (IL-4) for six days to generate monocyte-derived DC (MoDC). The Ag-primed MoDC are mixed with the Ag-stimulated mdPBMC and cultured for an additional six days. At six days, monocytes are isolated from blood and cultured to generate MoMΦ target cells. As described for analysis of the recall response, sets of doubly stimulated mdPBMC are co-cultured with infected target cells. After 24 h incubation, the culures are processed to analyze the CD8 CTL response and extent of killing of bacteria by CD8 CTL [46].

**Figure 3 vaccines-09-01085-f003:**
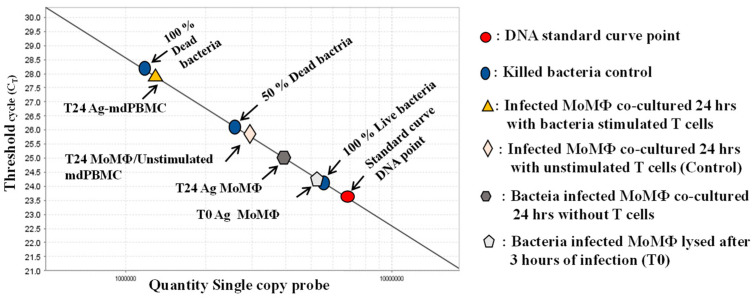
Illustration of the quantitative PCR method using propidium monoazide and a single copy gene from *Map*, F57, or *lpqT* (Rv1016c) from Mbv BCG in order to distinguish live from dead bacteria. A standard curve is generated from a single copy gene. A known concentration of DNA from live bacteria (100%), a mixture of 50% live bacteria and 50% dead bacteria, and 100% dead bacteria are used to show the range of sensitivity for quantifying the proportion of intracellular bacteria killed by CD8 CTL. Controls include the percent of live bacteria present in MoMΦ 3 h after uptake by MoMΦ, 24 h after uptake, and 24 h after incubation of MoMΦ with unstimulated mPBMC. Covalent binding of PMA to DNA is incomplete, allowing for a comparison of the differences in the percent of DNA from live and dead bacteria from infected MoMΦ target cells in relation to the DNA detected in the 100% live, 50% live and 50% dead, 100% dead control [46,51].

**Figure 4 vaccines-09-01085-f004:**
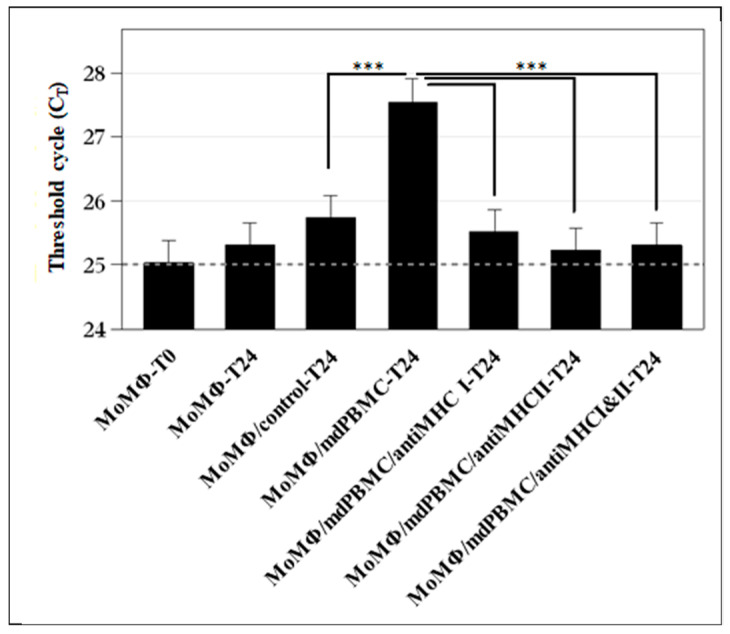
Histogram illustrating simultaneous interaction of antigen-primed antigen presenting cells (APC) with CD4 and CD8 T cells is essential for tri-directional signaling needed to elicit development of CD8 cytotoxic T cells. As illustrated in the figure, maximal killing of intracellular *Map* was observed with cultures of mdPBMC stimulated with MMP-primed APC in the absence of anti-MHC mAbs (compare center column with negative controls on the left). Development of CD8 CTL was interrupted in cultures of mdPBMC containing mAbs to MHC I and II, and cultures containing mAbs to MHC I or MHC II alone (compare the central column with results with cultures containing mAbs to MHC) on the right of the histogram plots. Data shown are the least squares means and standard deviations for experiments on blood collected from 3 steers. Significance symbols represent P-values adjusted for all pairwise comparisons such that: *** Padj < 0.001. See citations for methods and documentation [46,54].

## Data Availability

All the information and data referred to in this review is included in the original publications.

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
