# Peer review of "Advances in Understanding of the Immune Response to Mycobacterial Pathogens and Vaccines through Use of Cattle and Mycobacterium avium subsp. paratuberculosis as a Prototypic Mycobacterial Pathogen"

_vaccines, 2021, doi:10.3390/vaccines9101085_

Round 1

Reviewer 1 Report

  Bovine model is a good supplement to the current mice, humans model, and can be use to compare the difference between non-human primates and human primates. Cell mediated immune response plays major role in the vaccine. The authors did a great job in writing this review paper, but I have some suggestions listed as follows:

  1. Too many data listed in the paper and not organized clear. It would be better if you can organize the current trends in vaccine together, like gene-deleted vaccine, pipetide vaccine....; the vaccine mechanism  together,  don't mix vaccine and mechanism together.
  2. Line 30 Mtb, Mbv and Map should use full name, Line 37 and 38 can use abbreviation.

Author Response

We appreciate the time and effort the reviewer took to review the manuscript and make suggestions on how the manuscript could be improved. We agree there are a lot of data presented in the manuscript that were obtained over more than a 30 year span of time. Advances in understanding of the immune response and development of vaccines were made possible by taking advantage of technological advances made by other investigators and results obtained through collaborative studies. We have presented summaries of results chronologically, to show how we adapted technologies for use in the study of mycobacterial pathogens using cattle as the species of choice for conducting the studies. We have provided responses to the reviewer’s comments and suggestions below each comment.

Rev 1:  Bovine model is a good supplement to the current mice, humans model, and can be use to compare the difference between non-human primates and human primates. Cell mediated immune response plays major role in the vaccine. The authors did a great job in writing this review paper, but I have some suggestions listed as follows:

  1. Too many data listed in the paper and not organized clear. It would be better if you can organize the current trends in vaccine together, like gene-deleted vaccine, pipetide vaccine....; the vaccine mechanism  together,  don't mix vaccine and mechanism together. 

As mentioned, we have presented summaries of results at different stages of our studies in a chronological order to show how advances in knowledge on the immune response were made, starting with the development of monoclonal antibodies to bovine leukocytes differentiation molecules identical to those developed for studies in humans. This made it possible to use cattle as a model outbred species to extend studies on mycobacterial pathogens difficult to conduct using mice. We then describe results obtained through use of cattle. We summarize results showing infection with Map is typical of infection with other non-tuberculous mycobacterial pathogens that lead to establishment of a persistent infection controlled by an immune response that is unable to clear the infection. We developed the cannulated ileum calf model to make it possible to study the early and late events of infection in real time not possible using mice. Adaptation of methods for deleting genes provided opportunities to follow up on deletion of a gene, relA, in Mtb to show deletion had the same effect with Map and BCG indicating relA is the Achilles’ heel for mycobacterial pathogens. Development of methods to study the recall and primary immune responses to Map and BCG made it possible to show immunization ex vivo leads to development of CD8 cytotoxic T cells able to kill intracellular bacteria. Analysis of the immune response using the platforms showed relA deletion mutants are novel vaccine candidates and that further investigation of a membrane protein in Map is a more specific candidate vaccine with demonstrated effector CTL activity. So we have attempted to thread the major events of our studies into what we hoped would be an interesting informative review. In part as a result of suggestion made by this reviewer and the other 2 reviewers, we have added figures to illustrate the methods developed to conduct the studies and the major finding on the requirements for development of peptide based vaccines.

  1. Line 30 Mtb, Mbv and Map should use full name, Line 37 and 38 can use abbreviation.

We made the requested edit.

Reviewer 2 Report

This review article summarizes the progress of research on past vaccine development strategies for Mycobacterium avium subsp. paratuberculosis infection, which are still difficult to prevent.

Introduction: The intro, which briefly summarizes the history of the acquisition of pathogenicity of MAP and is interesting for general readers. Please also note that Johne and Frothingham found that tuberculosis and lesions were different from tuberculosis, including that tuberculosis in humans and cows was also rampant around this time.

L72: “2. Mycobacterium avium subsp paratuberculosis, a prototypic mycobacterial pathogen”
Isn't this title a little overstated? Isn't the prototype of acid-fast bacillus research the study of tubercle bacilli? As a veterinary researcher, I know.

L78-79: “The potential of Map as a zoonotic pathogen had been reported”
It's important, but the impression that it suddenly appears. It is not the purpose of this paper, but why not briefly describe here what was raised and what is currently up to date?

L87~: “A series of recommendations were made for the conduct of research that included.”
The reader expects the answers to the items 1) to 7) to be shown in this paper, but this is not the case. Is this all right?

L103~: 3. Immune response to Map
Each of the original studies presented here is voluminous and very long, but well organized and will help the reader's understanding.

L125~ 4. Early events of infection through direct infection of the ileum.
The description of the induction of immunity against MAP in the intestinal tract and mesenteric lymph nodes by the local infection experimental method introduced here is concise and good. How about writing that MAP invades from the dome M cells of the small intestine and everything starts?

L131: Discernable lesions were not detected until six months post-infection.
Does it means no granuloma formation for 6 months after infection. Why?

L148: Analysis of lymphocytes isolated----

Is it okay to understand that Mycobacterium avium japonicum, a complex immunological antigen complex, did not specifically stimulate lymphocytes in the small intestine? Is the immune response being raised masked? The author may write personal opinion.

L151: “Original isolates of Map from humans with CD elicit an immune response similar to the response elicited by isolates from cattle.”
It is said that the immune response evoked by the isolate from cattle and the Linda strain was similar, but it should be written in which animal the immunity was evoked. Cattle or mouse model?

L162: “Biopsies showed the bacteria were rapidly cleared from the ileum with no evidence of colonization throughout the eleven months of study.”

Infection experiments with bovine-derived strains indicate that the bacteria are eradicated in the early stages of infection, but what is the reason for the lack of evidence of growth in human-derived strains?

The CD8 T cell response was less in the early stages of infection, but is it related to the finding that it increased 11 months after the CD8 T cell response infection?

L225~: “In contrast, comparison of survival of a Map/relA deletion mutant with wildtype Map revealed deletion prevented establishment of persistent infection.”
It is very important results.

L267~:”This part is the most interesting one, but –“

 A lot of reported paper data has been introduced, but why not divide it into only the parts that are particularly important for beginners?

L292: It is good to mention important technical matters in experiments on infection and immunity of MAP, Such as fluorescent label by PMA, aggregation peculiar to Mycobacterium avium subsp.

L314: This part is also important to understand immunity against MAP, but the reader may need an easy-to-understand illustration.  

L344~: relA deletion mutants and Peptide-based vaccines and its method of delivery are written well.

 It may be possible to mention the effectiveness and problems of the MAP vaccine currently in use, which is behind the urgent need to develop a new vaccine.

The dormancy period in MAP infection seems to have important implications for the subsequent onset, but isn't it mentioned at the genetic level as well?

Author Response

We very much appreciate the time spent by this reviewer to review the manuscript and the time he/she? took to bring up areas of the presentation for discussion and potential revision. We have taken into account the suggestions of this reviewer and the other reviewers to improve the manuscript. We have written our replies below each of the reviewer’s comments.

This review article summarizes the progress of research on past vaccine development strategies for Mycobacterium avium subsp. paratuberculosis infection, which are still difficult to prevent.

Introduction: The intro, which briefly summarizes the history of the acquisition of pathogenicity of MAP and is interesting for general readers. Please also note that Johne and Frothingham found that tuberculosis and lesions were different from tuberculosis, including that tuberculosis in humans and cows was also rampant around this time.

We also recognize that Johne and Frothingham noted there were some notable differences in the lesions that develop from infection with Mtb and Mbv. This is of historical interest and could be a topic for discussion in a comparative review of early and contemporary studies of the histopathology of lesions that develop after infection with Mtb and Mbv.

L72: “2. Mycobacterium avium subsp paratuberculosis, a prototypic mycobacterial pathogen”
Isn't this title a little overstated? Isn't the prototype of acid-fast bacillus research the study of tubercle bacilli? As a veterinary researcher, I know.

 It is good to have discussions with a fellow veterinary investigator studying the pathogenesis of mycobacterial pathogens. We have attempted to write the review for a broader audience of investigators interested mycobacterial pathogens but pursuing studies of other pathogens. So we have used the title to specify the specific mycobacterial pathogen we used to advance studies on mycobacterial pathogens and the development of a bovine outbred species model for comparative studies of the immune response to mycobacterial pathogens.

L78-79: “The potential of Map as a zoonotic pathogen had been reported”
It's important, but the impression that it suddenly appears. It is not the purpose of this paper, but why not briefly describe here what was raised and what is currently up to date?

 We thought about this and how to present additional information. Since our review is more about the use of the bovine model to advance studies on the immune response to Map and other mycobacterial pathogens, we thought it was better to not expand on recent findings. Based on your query, we decided the best approach was to direct the readers to the most up to date information showing Map is the causative agent of CD. We have added a sentence directing them to citations that provide up to date information.

L87~: “A series of recommendations were made for the conduct of research that included.”
The reader expects the answers to the items 1) to 7) to be shown in this paper, but this is not the case. Is this all right?

A series of recommendations were made on where research should be conducted to better understand the mechanisms of pathogenesis of paratuberculosis. We believed it was essential to list most of the recommendations and then indicate, our studies focused on response to 2 of the recommendations. We indicated which recommendations we took up in response to the committee’s recommendations.

L103~: 3. Immune response to Map
Each of the original studies presented here is voluminous and very long, but well organized and will help the reader's understanding.

 We agree. It was very difficult to summarize the results from extensive studies conducted over some 30 years. We did are best to summarize the results and make the findings interesting to read about.

L125~ 4. Early events of infection through direct infection of the ileum.
The description of the induction of immunity against MAP in the intestinal tract and mesenteric lymph nodes by the local infection experimental method introduced here is concise and good. How about writing that MAP invades from the dome M cells of the small intestine and everything starts?

 We thought about how much information to add and thought it best not to digress and provide too much additional information. The object was to show we developed a less invasive method for continuous access to the ileum to study the interaction of Map with the ileal tissues during the early and late phases of infection with Map, events not possible to study in humans or mice. We stayed away from providing more specific detail on tracking the uptake of Map by different cell types. We would have only been citing studies by others and not any new information obtained by serial biopsies after infusion of bacteria into the ileum.

L131: Discernable lesions were not detected until six months post-infection.
Does it means no granuloma formation for 6 months after infection. Why?

There were some surprises when we were able to make and evaluate serial biopsies following infusion of bacteria. One of the observations was the rapid clearance of bacteria from ileal tissues. We thought we should be able to detect the presence of bacteria in some tissues for a few days and potentially transient appearance of some inflammation and at least transient appearance of lesions early and then persistent lesions/granulomas. We demonstrated we could cannulate cows at the late stage of infection but weren’t able to do any serial biopsies.

L148: Analysis of lymphocytes isolated----

Is it okay to understand that Mycobacterium avium japonicum, a complex immunological antigen complex, did not specifically stimulate lymphocytes in the small intestine? Is the immune response being raised masked? The author may write personal opinion.

Always some surprises when having more mAb reagents to conduct studies to characterize the cellular response immune response. We were looking at the early stages of infection with Map not knowing what we would find. We were studying the early stages of infection in real time. We didn’t observe any macroscopic or microscopic events of infection and inflammation. Consistent with these observations, we didn’t observe any clear evidence of an increase in expression of activation molecule on lymphocyte like increase in expression of the interleukin 2 (CD25). molecule on cells collected from ileal tissues at necropsy. We were looking at the early stages of infection so our observations showed there were no clear changes as detected by flow cytometry. No funds were available for conducting a more comprehensive study.

L151: “Original isolates of Map from humans with CD elicit an immune response similar to the response elicited by isolates from cattle.”
It is said that the immune response evoked by the isolate from cattle and the Linda strain was similar, but it should be written in which animal the immunity was evoked. Cattle or mouse model?

We added figures as a result of suggestions by one of the other reviewers that answer this potential misconception as to which species was used to obtain the information. We didn’t do any studies with mice. Our objective was to obtain relevant information from studies with cattle.

L162: “Biopsies showed the bacteria were rapidly cleared from the ileum with no evidence of colonization throughout the eleven months of study.”

Infection experiments with bovine-derived strains indicate that the bacteria are eradicated in the early stages of infection, but what is the reason for the lack of evidence of growth in human-derived strains?

We have to infer what the early stages of infection are in humans. The indwelling cannula provided an opportunity to observe whether the isolates obtained from humans had changed their capacity to infect and elicit cellular changes that differ from those observed with bovine isolates. We could only report what we observed. Like in cattle, infection is brought under immune control during the early stages of infection. This is what we observed with the isolate from humans and cattle. It would have been really interesting if we could have conducted a more extensive study. But we obtained the important information with the study.

The CD8 T cell response was less in the early stages of infection, but is it related to the finding that it increased 11 months after the CD8 T cell response infection?

We weren't able to pursue further comparative studies. But the studies suggested to us that the delay in a robust response could indicate this delay might contribute to the capacity of Map to establish a persistent infection.

L225~: “In contrast, comparison of survival of a Map/relA deletion mutant with wildtype Map revealed deletion prevented establishment of persistent infection.”
It is very important results.

This has been a real important observation to follow up on and this is what we are doing.

L267~:”This part is the most interesting one, but –“

 A lot of reported paper data has been introduced, but why not divide it into only the parts that are particularly important for beginners?

We wrote the review to try and include investigators just beginning to study immunopathogenesis of mycobacterial infection as well as seasoned investigators. We attempted to provide sufficient information and relevant references to bring everyone up to date on what we have learned thus far. We tried to pique interest in the observation that led us to investigate the initial observation. We used the next section of the paper to report further findings.

L292: It is good to mention important technical matters in experiments on infection and immunity of MAP, Such as fluorescent label by PMA, aggregation peculiar to Mycobacterium avium subsp.

We added a figure to illustrate the use of PMA with single copies of genes in Map and BCG to quantitate killing of intracellular bacteria using quantitative PCR.  A visual presentation does help understand how PMA is used.

L314: This part is also important to understand immunity against MAP, but the reader may need an easy-to-understand illustration.  

We added a figure to illustrate the use of PMA and quantitative PCR to study the killing of intracellular bacteria.

L344~: relA deletion mutants and Peptide-based vaccines and its method of delivery are written well.

 It may be possible to mention the effectiveness and problems of the MAP vaccine currently in use, which is behind the urgent need to develop a new vaccine.

The dormancy period in MAP infection seems to have important implications for the subsequent onset, but isn't it mentioned at the genetic level as well?

We tried to keep some of these thoughts in the minds of the audience as we described how we took advantage of the platform we developed to study and answer an unresolved question as to when and where signaling had to occur between the primed APC and CD4 and CD8 T cells. We added a figure used in our study to graphically illustrate the results of the study.

One of the historical misconceptions about the pathogenesis of Map (and actually the same thought for other mycobacterial pathogens) is the apparent dormant phase of the disease. It appeared, without experimental evidence, that the dormant of latency period that occurs after infection that Map had found a safe haven sequestered and hidden from the immune system. It was also thought that Map could pass through animals without eliciting an immune response because of the latency. Our initial studies showed exposure leads to 100% infection after exposure with no age difference. Our next studies showed an immune response develops immediately after infection that controls the infection but is unable to clear the infection leading to the development of the latent (apparent dormant) stage pf infection. We hade to improve the assays to demonstrate a cellular immune response can be detected within 14 days after experimental infection. This will be reported when we complete our ongoing studies.

I truly appreciate having an opportunity to discuss our studies with you and add information that we couldn’t put in the review.

Reviewer 3 Report

The manuscript submitted by Davis et al. entitled "Advances in understanding of the immune response to mycobacterial pathogens and vaccines through use of cattle and Mycobacterium avium subsp paratuberculosis as a prototypic mycobacterial pathogen" deeply reviews the current knowledge about the pathogenesis,  virulence and modulation of immune response in infection by Mycobacterium tuberculosis, Mycobacterium bovis and Mycobacterium avium subsp paratuberculosis. The content of the manuscript is relevant to the field and is very well written. However, the text in the lines 88-93 should be reviewed regarding the punctuation. Additionally, the authors should provide 1 or 2 images along the manuscript to capture more the interest of the reader. Finally, a phylogenetic study of the aminoacidic sequences of the major virulence factors between the 3 pathogenic agents will also improve the good quality of the manuscript.

Author Response

We appreciate the time and effort of the reviewer to carefully review the manuscript and offer suggestions on how to improve the content.

In response to first comment on punctuation, we have made the appropriate corrections.

In response to the second comment and suggestion, we have added figures to make the presentation more interesting and to provide more detail on the methods developed to conduct the studies. We had thought about adding some figures. We were hoping one of the reviewers would make a suggestion that we could respond to.

Round 2

Reviewer 3 Report

The authors positively answered to all the questions raised by this reviewer. The manuscript is suitable for publication in the present form.